# Molecular Tools to Infer Resistance-Breaking Abilities of Rice Yellow Mottle Virus Isolates

**DOI:** 10.3390/v15040959

**Published:** 2023-04-13

**Authors:** Laurence Dossou, Agnès Pinel-Galzi, Jamel Aribi, Nils Poulicard, Laurence Albar, Sorho Fatogoma, Marie Noëlle Ndjiondjop, Daouda Koné, Eugénie Hébrard

**Affiliations:** 1AfricaRice Center, M’bé Research Station, Bouaké 01 BP 2551, Côte d’Ivoire; laurencedossou@yahoo.com (L.D.); m.ndjiondjop@cgiar.org (M.N.N.); 2WASCAL/CEA-CCBAD, Université Félix Houphouët-Boigny, Abidjan 01 BP V 34, Côte d’Ivoire; fsorho@gmail.com (S.F.); daoudakone2013@gmail.com (D.K.); 3PHIM, Plant Health Institute, University Montpellier, IRD, INRAE, CIRAD, SupAgro, 911 Avenue Agropolis, 34394 Montpellier, France; agnes.pinel@ird.fr (A.P.-G.); jamel.aribi@ird.fr (J.A.); nils.poulicard@ird.fr (N.P.); laurence.albar@ird.fr (L.A.)

**Keywords:** RYMV, rice, resistance, virulence, detection

## Abstract

Rice yellow mottle virus (RYMV) is a major biotic constraint to rice cultivation in Africa. RYMV shows a high genetic diversity. Viral lineages were defined according to the coat protein (CP) phylogeny. Varietal selection is considered as the most efficient way to manage RYMV. Sources of high resistance were identified mostly in accessions of the African rice species, *Oryza glaberrima*. Emergence of resistance-breaking (RB) genotypes was observed in controlled conditions. The RB ability was highly contrasted, depending on the resistance sources and on the RYMV lineages. A molecular marker linked to the adaptation to susceptible and resistant *O. glaberrima* was identified in the viral protein genome-linked (VPg). By contrast, as no molecular method was available to identify the hypervirulent lineage able to overcome all known resistance sources, plant inoculation assays were still required. Here, we designed specific RT-PCR primers to infer the RB abilities of RYMV isolates without greenhouse experiments or sequencing steps. These primers were tested and validated on 52 isolates, representative of RYMV genetic diversity. The molecular tools described in this study will contribute to optimizing the deployment strategy of resistant lines, considering the RYMV lineages identified in fields and their potential adaptability.

## 1. Introduction

Rice yellow mottle virus (RYMV) is the most damageable rice virus in Africa [1]. RYMV is a member of the genus *Sobemovirus* in the family *Solemoviridae* [2]. RYMV has non-enveloped monopartite icosaedric particles and single-stranded messenger-sense RNA genome of ca. 4450 nucleotide long. The RYMV genome is organized in five overlapping open reading frames (ORFs). Detected in more than 25 rice-growing countries, severe RYMV epidemics caused high yield losses [3]. Early symptoms are yellow mottle on leaves and dwarfing of plants. The disease can lead to grain sterility, drying out, and plant death. The host range of RYMV is narrow, and it is restricted to rice and a few wild grasses. Highly adapted to its plant host, RYMV multiplies rapidly and spreads widely in the plant tissues. High virus content in plants facilitates mechanical transmission by contact and/or through chewing insects, such as beetles. Transmission of RYMV involves multiple biotic and abiotic vectors, including human, as well as agricultural practices, such as rice transplantation [4]. These biological characteristics make field prophylaxis difficult.

Varietal selection is a promising management strategy. Highly resistant accessions, which are characterized by a lack of symptoms, very low to undetectable virus content and no yield loss upon infection have been identified [5,6]. Only two highly resistant accessions, namely, Gigante and Bekarosaka, belong to *Oryza sativa indica*, the Asian rice species, widely cultivated due to its high yield potential. Although the high resistance of Gigante has been successfully transferred in high yielding varieties [7,8], these varieties have not been widely deployed yet in the field. The other resistance sources come from the African rice species, *O. glaberrima*, which is currently much less cultivated in Africa. Their introgression in *O. sativa* varieties is complicated by an interspecific sterility barrier between the two species [9]. Three resistance genes, *rymv1*, *rymv2,* and *RYMV3,* have been described, and all known resistant accessions to date carry a resistance allele on at least one of them, suggesting that the chances of identifying new resistance genes in the diversity of cultivated rice are very low [10]. In this context, even before resistance breeding allows the deployment of all known resistance genes, optimizing resistance sustainability is a major challenge.

Experimental evolution on resistant accessions has been carried out under controlled conditions with hundreds of RYMV isolates from all over Africa [9,10,11,12]. These studies revealed that the frequency of resistance-breaking (RB) events depends on both resistant accessions and viral lineages. The *rymv2* gene, coding for a nucleoporin (CPR5-1), is rapidly overcome by half of the isolates tested [11,12]. RB mutations emerged in the putative membrane anchor domain of the viral polyprotein P2a, but the molecular mechanism involved is still unknown. The *rymv1* and *RYMV3* genes appeared to be less frequently overcome than *rymv2*, but some RYMV isolates are able to adapt and overcome them. *RYMV3* encodes a nucleotide-binding and leucine-rich repeat domain protein (NLR) involved in the plant immune response, and resistance breakdown occurs through mutations in the viral coat protein (CP) that prevent its recognition by the NLR [13,14]. The durability of *rymv1*-mediated resistance has been deeply investigated. The *rymv1* gene encodes an isoform of the eukaryotic translation initiation factor 4G, eIF(iso)4G1, a component of the host translation complex. Recruited by several viral families, these host proteins represent major susceptibility factors [15]. Rice resistance is associated with the defect in eIF(iso)4G1 interaction with the viral genome-linked protein (VPg) of RYMV. Mutations in VPg, a major virulence and adaptation factor, restore the interaction and lead to resistance-breakdown [16,17]. Alternative mutational pathways outside VPg, in the C-terminal domain of the P2a, have also been rarely observed, but the molecular mechanism involved is still unknown [18].

RYMV lineages showed highly contrasted RB abilities. RB mutations have rarely emerged in East African virus lineages (strains S4–S6), except to overcome the resistance of *O. sativa* accessions carrying the *rymv1–2* allele [19]. In contrast, lineages located in West Africa (strain S1–S3) have a higher propensity to adapt to resistant accessions of *O. glaberrima*, regardless of the resistance genes and alleles they carry [12,13,14,19,20]. Lineage adaptabilities reflect their evolutionary history on rice species during the dispersal of the RYMV from East to West Africa [21]. The preadaptation of strains S1–S3 to susceptible *O. glaberrima* accessions, historically cultivated in West Africa, increases their ability to overcome resistance carried by *O. glaberrima* accessions. A major molecular signature of this ancient adaptation has been identified at position 49 of VPg. A glutamic acid (E) at position VPg49 in East African lineages evolved to a threonine (T) in some West African lineages. The impact of VPg49T was experimentally demonstrated both in RYMV isolates and mutated infectious clones, increasing both the viral fitness on the susceptible *O. glaberrima* and the viral RB ability on the resistant *O. glaberrima* [21]. Based on their amino acid at position VPg49 and their contrasted *rymv1*-RB ability, two RYMV pathotypes, named E and T, have been defined [19,20]. Later, Hébrard et al. [22] showed that a subset of isolates from West-Central Africa (strain S1ca), with the threonine on VPg49, are able to adapt at a high frequency to all known highly resistant accessions, from both *O. sativa* and *O. glaberrima* species. These isolates defined a new pathotype, named hypervirulent T’. 

Up to now, plant inoculation assays were the only method to determine RYMV pathotypes [12,13,19,20,22]. A time-consuming experiment is required to inoculate each isolate on at least 20 plants of each of the six reference resistant accessions in controlled conditions and to detect the emergence of RB genotypes using DAS-ELISA up to two months after inoculation. An alternative method was to infer the pathotypes by partial sequencing of the CP and VPg genes to determine the RYMV strain (S1–S3 or S4–S6) and the polymorphism at position 49 (glutamic acid E or threonine T), respectively. However, no molecular marker was available to identify the hypervirulent lineage, and sequencing increased the lab expenses. In this study, based on previous data on the phylogeny of the RYMV genome and resistance-breaking experiments, (i) a specific molecular signature of the hypervirulent T’ pathotype was identified, (ii) RT-PCR primers specific to the three pathotypes were designed to easily infer the RB ability of a given isolate, and (iii) these molecular tools were validated on 52 isolates, representative of the genetic diversity of RYMV in Africa and Madagascar.

## 2. Materials and Methods

### 2.1. Viral Isolates, Pathotypes and Sequences

A total of 88 RYMV isolates, representative of the RYMV genetic and pathogenic diversity, were analyzed in this study (Appendix A). The names of the isolates are composed of two letters, depending on their country of origin and a number.

The full-length sequence of 67 isolates has been previously published [23,24], including 37 originating from West Africa, 24 from East Africa, and six from Madagascar. For the 21 other isolates, the CP and VPg sequences were already available in the NCBI database or obtained in this study (Appendix A).

Fifty-two isolates were selected from previous collections of infected rice leaves stored at −20 °C or −80 °C. The dataset contained 17 isolates previously pathotyped [22], including six isolates from the pathotype E, six from the pathotype T, and five from the pathotype T’. Twelve out of 17 pathotyped isolates have been previously sequenced. For the other 35 isolates, their pathotypes were inferred using the molecular tools designed in this study. Nineteen out of 35 have been entirely sequenced, and 16 were only CP sequenced.

### 2.2. Sequence Analyses, Polymorphism Identification and Primer Design

A phylogenetic analysis was performed using 67 full-length sequences. Sequences were aligned using CLUSTAL-W [25], implemented in MEGA6 [26], with default parameters. Maximum-likelihood (ML) phylogenetic trees were reconstructed using the PHYML-3.1 algorithm implemented in SEAVIEW v4.7 [27] under the best-fitted nucleotide substitution model (GTR+G+I). A phylogenetic tree was drawn using FigTree, v1.3.1 (http://tree.bio.ed.ac.uk/software/figtree/ accessed on 4 March 2023).

Polymorphism analysis was carried out using DNAsp v6.12.03 [28]. To identify pathotype-specific polymorphisms, the sequences of the sub-lineage in which pathotype T’ has been identified were defined manually as a population, and DNA divergence from the rest of the sequences was calculated. Polymorphisms at the sites differentially occupied by specific nucleotides in the sub-lineage were identified (Table 1).

The conservation of nucleotides surrounding the candidate pathotype-specific positions were examined from the alignment of the genomic sequences. Primers were designed with Seqbuilder pro6 (DNAstar) and ordered to Eurogentec (Seraing, Belgium).

### 2.3. RNA Extractions and RT-PCR Amplifications 

Total RNA of RYMV-infected leaves was extracted from the 52 isolates using the GeneJET Plant RNA Purification Kit (ThermoFisher Scientific, Waltham, MA, USA). Two independent reverse transcription reactions were performed with 2365R and R4 reverse primers at 100 mM (Table 2). After a denaturation step at 70 °C for 5 min, dNTPs, RNAsin, and MMLV-reverse transcriptase (RT) were added to the reaction mixture in the appropriate buffer (Promega, Madison, WI, USA) and incubated at 42 °C for 1 h. Two polymerase chain reactions for each RT reaction were prepared with 2 µL of cDNA products, 0.4 µM of reverse and forward primers, dNTPs, and Taq’Ozyme HS PCR enzyme (Ozyme, Paris, France). For the PCR reactions with 2365R and F1731E or F1731T, the reaction mixtures were subjected to denaturation at 94 °C for 5 min, 28 cycles of 1 min at 94 °C, 30 s at 63 °C, and 1 min at 72 °C, and then they were incubated at 72 °C for 10 min and stored at 15 °C. For the PCR reactions with R4 and F353T or F353G, the reaction mixtures were subjected to denaturation at 94 °C for 3 min, 25 cycles of 30 s at 94 °C, 30 s at 67 °C, and 1 min at 72 °C, and then they were incubated at 72 °C for 5 min and stored at 15 °C. Each RT-PCR was repeated two or three times independently. The PCR products were loaded and visualized by the electrophoresis of ethidium bromide stained 1% agarose gel in buffer TBE 0.5X at 100V for 20 min.

In addition, the VPg gene of 16 isolates was amplified by RT-PCR using the forward primer F1SNP 5′-CCCGCTCTACCACAA-3′ and the reverse primer R14bis 5′-ACTTCGCCGGTTTCGCAGAGGATT-3′ [29,30] and sequenced to complete the dataset. Briefly, the reaction mixtures were subjected to denaturation at 94 °C for 3 min, 30 cycles of 30 s at 94 °C, 30 s at 54 °C, and 1 min at 72 °C, and then they were incubated at 72 °C for 10 min and stored at 15 °C. Sanger sequencing was subcontracted to Genewiz (Azenta Life Sciences, Leipzig, Germany).

## 3. Results

### 3.1. Identification of Polymorphisms Specific of E/T/T’ Pathotypes

RYMV phylogeny was reconstructed using 67 full-length sequences from 18 countries in Africa (Figure 1, left panel). As previously described, West African lineages (strains S1–S3 and minor lineages Sg and Sa) are monophyletic, contrary to East African ones (strains S4–S6 and the associated lineages). Based on the published data [19,20,22], pathotype E is found in each major strain, pathotype T is limited to West African lineages, and pathotype T’ is restricted to a sub-lineage of the West-Central African strain S1ca.

**Figure 1 viruses-15-00959-f001:**
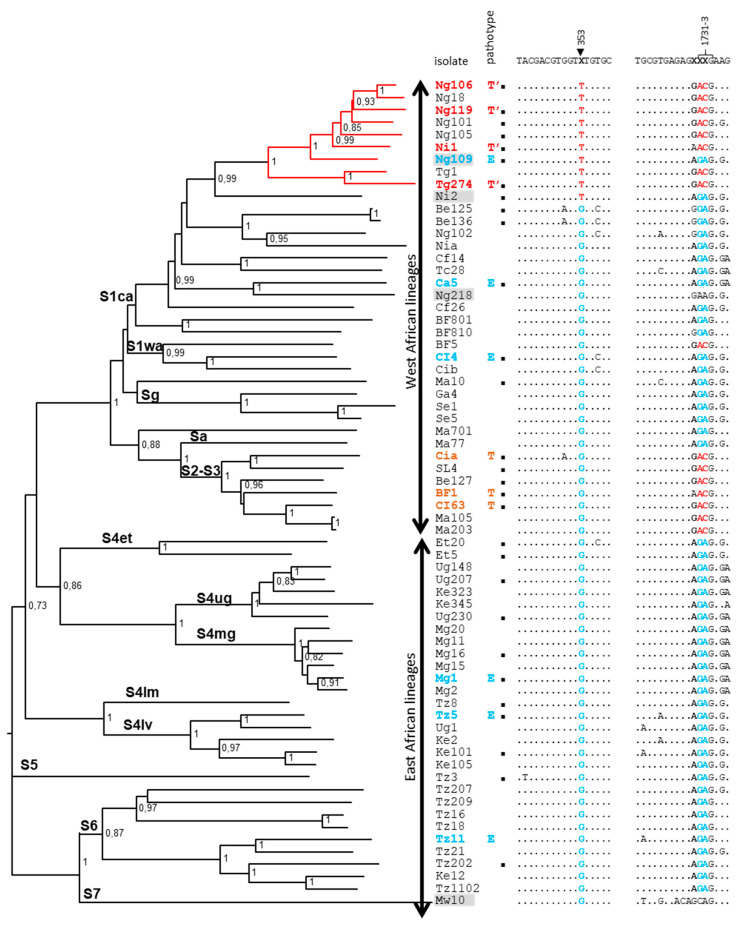
(**Left panel**) Phylogenetic tree of 67 genomic sequences representative of RYMV genetic diversity in Africa. Strains (S1–S7), lineages, and bootstraps were indicated on the tree. The lineage with T’ isolates is colored in red. Resistance-breaking ability is indicated by bolded isolate names (blue, orange, and red for pathotypes E, T, and T’, respectively). (**Right panel**), Partial alignments at polymorphic domains chosen to design specific primers. Specific polymorphic sites used for specific primers were highlighted in red or blue. Genomic numbering was indicated based on the reference isolate CIa. Fully sequenced isolates used to validate the molecular tools designed in this study are indicated by black squares. Isolates excluded from the sequence analysis were highlighted in grey.

Sequences were ordered in the alignment with respect to the phylogenetic tree to better visualize clustering of polymorphisms (Figure 1, right panel). As previously described, the codon at position 49 of the VPg, located at nucleotides 1731–1733 of the genomic sequence, is a molecular signature that distinguishes the E pathotype, characterized by the codon GAG (=glutamic acid, E), from the T and T’ pathotypes, characterized by the codon ACG (=threonine, T). Two non-canonical VPg49 codons, AAG (=lysine, K) and CAG (=glutamine, Q), for isolates Ng218 and Mw10, respectively, were found in our dataset. The presence of VPg49Q in the minor strain S7, a strain restricted to southern Malawi, has been previously described [31]. The isolate Ng218 was further investigated. As the initial isolate was not available anymore, sequencing was performed on a sample from a previous back-inoculation, but the presence of VPg49K was not confirmed (codon occupied by GAG, i.e., 49E). These two isolates were excluded from the following analyses.

To identify a molecular signature of the pathotype T’, we considered that the four other isolates that were full-length sequenced (Ng18, Ng101, Ng105, Tg1) from the same sub-lineage as the four pathotyped isolates (Ng106, Ng119, Ni1, Tg274) also belonged to this pathotype (Figure 1, left panel, red lineage). The isolates Ni2 and Ng109 were excluded from the following analyses. Indeed, the pathotype of Ni2 is still unknown due to the failure of multiplication of the initial sample on susceptible rice, and the pathotype of Ng109 was previously identified as E, but a recombination signal was detected in its sequence (N. Poulicard, unpublished data).

DNAsp polymorphism analysis was carried out using this dataset made of the 63 RYMV genomic sequences. All the isolates from the sub-lineage of pathotype T’ were grouped in a population, and DNA divergence from the rest of the sequences was calculated. Sites differentially occupied by specific nucleotides were identified, and polymorphisms at these sites were further analyzed, segregating the isolates, depending on their strains and on the codon VPg49 (codon ACG/GAG at position 1731–1733) (Table 1). Only three sites along the genome, at positions 353, 673, and 2539, have fixed differences between the sub-lineage of T’ isolates and all the other isolates. Two other sites, 1772 and 3198, were monomorphic in the sub-lineage of T’ isolates, but they were polymorphic in the other sequences, which is not efficient to design specific primers. Sites 2863, 3301, and 3343 have fixed differences between the sub-lineage T’ and the other isolates from the data set of 63 sequences. However, this was not confirmed by combining preliminary data on other ongoing datasets from West and East Africa, (personal communications, D. Fargette and N. Poulicard), so these three sites are not adapted to design specific primers. Altogether, only the polymorphisms G353T, A673G, and C2539T, located, respectively, in P1, Pro, and putative P10/P8 genes, are specific of the sub-lineage of pathotype T’.

### 3.2. Design of the Pathotype Inference Assay

Firstly, the nucleotides surrounding position 49 of the VPg (nts 1731–1733) were analyzed in order to design primers that are specific of the E vs. T/T’ isolates (Figure 1, right panel). As previously published, the changes at codon 49 are coordinated with those at the adjacent codon 48 and, to a lesser extent, codon 50 [19,21]. The basal motif found in strains S4–S6 is AG**A**-**GA**G-A**G**G (=R-E-R), while the derived motif found in some lineages of strains S1–S3 is AG**G**-**AC**G-A**A**G (=R-T-K). Some exceptions were observed in strain S6 with the R-E-K motif [19]. Based on these data, we designed two forward primers, named F1731E and F1731T, able to discriminate E/T polymorphism at positions 1731–1733, using only the third nucleotide of codon 48 and the two first nucleotides of codon 49 (Table 2, Figure 2). After a RT reaction with the common reverse primer 2365R, the forward primers were added to perform independently two PCR reactions. A PCR band at 638 pb is specifically amplified, with one or the other forward primer, depending on the E/T polymorphism.

Secondly, the nucleotides surrounded positions 353, 673, and 2539, which were analyzed in order to design specific primers. Two and four polymorphic sites were detected in the close vicinity of the candidate sites 673 and 2539 (Appendix A). As these sites vary independently of the A/G and C/T polymorphisms at positions 673 and 2539, respectively, they impaired the design of pathotype-specific primers targeting these regions. By contrast, the nucleotides surrounding the site 353 are highly conserved, and only one polymorphism was observed in three isolates at position 350 (Figure 1). Two forward primers, named F353T and F353G, were designed with, respectively, T or G at their 3′ extremity at position 353 (Table 2). After a RT reaction with the common reverse primer R4, the F353T and F353G forward primers were added in the tube to perform, independently, two PCR reactions. A PCR band at 280 pb is specifically amplified for T/T’ pathotypes, respectively (Figure 2).

### 3.3. Validation of the Molecular Tools

Fifty-two isolates were selected to be representative of the RYMV genetic and pathogenic diversity (Appendix A). The sample set contained 17 isolates that were previously pathotyped [22], including six isolates from the pathotype E, six from the pathotype T, and five from the pathotype T’. Twelve out of 17 pathotyped isolates have been previously sequenced. For the other 35 isolates, their pathotype were inferred using our molecular tools based on the polymorphisms at positions 353 and 1731–1733. Nineteen out of 35 have been entirely sequenced and 16 only CP-sequenced.

In a first step, the isolates were analyzed with F1731E/2365R and F1731T/2365R primer pairs to infer the VPg49 polymorphism and the pathotype E vs. T/T’ (Figure 3 left panel). Thirty-five out of 52 isolates tested (67%) gave the expected profile with one band at 638 bp in only one of the two reactions. Two other profiles were observed for 13 isolates (25%), with a strong band with one pair and a feint band with the other (Appendix A). Nevertheless, the repeatable and high quantitative difference between the two reactions can be used to infer the pathotype. The VPg49 polymorphism inferred from the RT-PCR assay was compared to the published sequences, and new sequencing data were produced in this study. In total, the VPg49 polymorphism inferred with the F1731E/F1731T primers was validated on 48/52 isolates (92%) (Table 3). Four isolates (Ni2, Be136, and Ca54, strain S1ca, VPg49E and CI1, strain S1wa, VPg49T) gave undetermined profiles with two bands at the same intensity with the two PCR reactions (Appendix A).

In a second step, the 26 RYMV isolates with a profile T/T’ were analyzed with F353T/R4 and F353G/R4 primer pairs. The two expected profiles, giving only one band at 280 bp in one of the two reactions, were detected for 25/26 isolates (96%) (Figure 3 right panel, Table 4). By contrast to the first step of the pathotype inference step, the second step is qualitative (+/−). Only one isolate (SL3, strain S3) gave an undetermined profile characterized by the absence of amplification both with F353T/R4 and F353G/R4 primers, while the PCR with F1731T/2365R primers was positive. The polymorphism G/T at position 353 inferred from the RT-PCR assay was compared to the published data. Twelve sequences out of 25 are available and 100% of the inferences were validated. Considering that pathotype T’ is restricted to one sub-lineage, the discrimination of T/T’ pathotypes was based on phylogenetic analysis and again 100% of the molecular inferences were validated.

## 4. Discussion

In this study, RT-PCR primers were designed to easily infer E/T/T’ pathotypes and the RB abilities of RYMV isolates avoiding the pathotyping and sequencing steps. Whether many powerful methodologies have been described for SNP genotyping [32], they frequently rely on fluorescence analyses or complex detection methods (such as real time-quantitative RT-PCR, microarrays, or high-throughput sequencing), which are not always available near RYMV sampling areas. Other methods are based on restriction site specificity, but they largely depend on the sequence of interest, and this strategy failed to detect E/T polymorphisms. The methodology developed here was based on the primer extension approach with detection of PCR product on gel electrophoresis, which can be achieved with basic laboratory equipment and consumables. These molecular tools were validated on 52 isolates representative of the RYMV genetic and pathogenic diversity.

The F1731E/T primers were based on the polymorphism VPg49E/T, previously demonstrated as involved in RYMV evolution and adaptation to susceptible and resistant *Oryza glaberrima*. Other specific primers to detect rare alternatives at this position, such as VPg49Q, found in the minor strain S7, were not investigated. Despite several optimization steps, including an increase in the hybridization temperature, quantitative differences between RT-PCR fragments have to be considered in 25% of cases to determine putative profiles E vs. T/T’. The addition of modified nucleotides [33] or punctual mismatches [34] would be a promising pathway to prevent the occurrence of unspecific bands. Regarding the undetermined profiles (isolates Ni2, Be136, Ca54 and CI1), viral genotypes were verified by Sanger resequencing of the VPg and CP genes. Double pics were detected in the electropherograms of the isolate Ni2, explaining its atypical profile. The presence of mixed infections at low level would not be excluded for the three remaining exceptions.

The F353T/G primers were designed after polymorphism analysis of the RYMV genome dataset applying high stringency. This simple method resulted in the identification of the first molecular marker of the lineage in which pathotype T’ has been identified. Despite the fact that these primers have only one base different, they are highly efficient. No data are available yet on the direct or indirect role of 353G in the remarkable RB ability of this lineage. This polymorphism is silencious, and it is located at the third position of codon 91 of the P1. The hypothesis of a neutral marker cannot be excluded. Up to now, isolates of the pathotype T’ are rare and genetically related. Complementary surveys in West Africa, including Niger, Togo, Nigeria, Benin, and Burkina Faso, would help to better assess the geographical distribution of the hypervirulent pathotype T’ and the genetic structure of the corresponding sub-lineage.

Accuracy of the molecular tools designed in this study was evaluated for each pathotype through calculation of type I and type II errors [35,36] (Appendix A). For pathotype T’, the ability to avoid false positives, referred as specificity (type I error), and the ability to avoid false negatives, i.e., sensitivity (type II error), were both 100%. For pathotypes E and T, specificity and sensibility were comprised between 88% and 96% due to undetermined profiles found in four isolates with F1731E/T primers. 

In conclusion, the protocol set up in this study provides a simple, fast, and accessible tool for breeders and pathologists to infer the RB abilities of newly surveyed RYMV isolates. However, phenotyping assays in controlled conditions will be needed to confirm the durability of resistance genes introgressed in newly improved rice lines. In addition, monitoring of RB emergence in field conditions should be reinforced in particular in high risk areas of West Central Africa. Used at a large scale, our method would contribute to optimize the deployment strategy of resistant lines considering the RYMV lineages identified in fields and their potential adaptability.

## Figures and Tables

**Figure 2 viruses-15-00959-f002:**
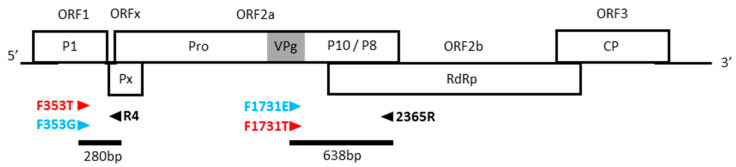
Positions of primers on RYMV genomic organization. Open reading frames (ORFs) and the proteins they encoded are indicated. Protein P1, protein X (Px), proteinase (Pro), VPg (in grey), putative P10/P8, RNA-dependent RNA-polymerase (RdRp), and coat protein (CP). Primer positions and PCR products are indicated by triangles and solid lines, respectively.

**Figure 3 viruses-15-00959-f003:**
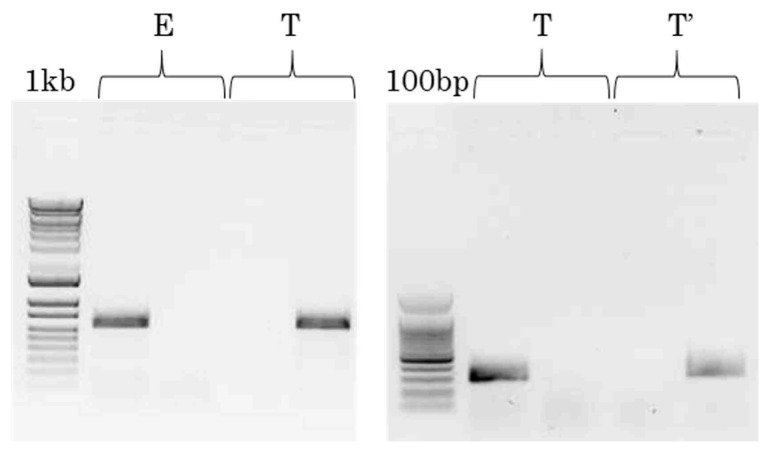
(**Left**
**panel**). Major RT-PCR profiles obtained using F1731E/2365R (lines 2 and 4) and F1731T/2365R (lines 3 and 5) primer pairs in two independent reactions for each RYMV isolate. Line 1, size marker (1kb); lines 2–3, isolate Ma2, profile E; lines 4–5, isolate Ma105, profile T. (**Right**
**panel**). Major RT-PCR profiles obtained using F353T/R4 (lines 2 and 4) and F353G/R4 (lines 3 and 5) primer pairs in two independent reactions for each T/T’ isolate. Line 1, size marker (100 base pairs); lines 2–3, isolate CI2, profile T; lines 4–5, isolate Tg247, profile T’.

**Table 1 viruses-15-00959-t001:** Nucleotidic polymorphisms specific to E/T/T’ pathotypes. For each strain, N isolates were grouped, depending on their polymorphisms at positions 1731–1733 (codons ACG or GAG), corresponding to VPg49 (glutamic acid E or threonine T, respectively). Nucleotides at the candidate sites identified by DNAsp were indicated. Polymorphic sites specific to pathotypes were bolded and colored in red (T or T’ specificity) or blue (E specificity).

		VPg49	P1	Pro	VPg	P10/P8	RdRp			
strain	N	1731-3	353	673	1772	2539	2863	3198	3301	3343
S1ca	8	**AC**G**=T**	**T**	**G**	A	**T**	A	A	A	T
	10	**GA**G**=E**	**G**	**A**	T	**C**	G	C	G	C
S1wa	1	**AC**G**=T**	**G**	**A**	T	**C**	G	C	G	C
	3	**GA**G**=E**	**G**	**A**	T	**C**	G	C	G/A *	C
Sg	3	**GA**G**=E**	**G**	**A**	T	**C**	G	C	G	C
Sa	2	**GA**G**=E**	**G**	**A**	T	**C**	G	C	G	C
S2–S3	7	**AC**G**=T**	**G**	**A**	T	**C**	G	C	G/A *	C
all S4	19	**GA**G**=E**	**G**	**A**	T/C	**C**	G/A *	C	G/A *	C/T *
S5–S6	10	**GA**G**=E**	**G**	**A**	T/C	**C**	G	C/T	G	C

* Polymorphisms reported in personal communications, D. Fargette and N. Poulicard.

**Table 2 viruses-15-00959-t002:** PCR primers used to discriminate E/T/T’ pathotypes. Names of PCR primers and specific nucleotides are colored according Figure 1.

Name	Sequence	Reference
F1731E	5′ACC-TGG-GTG-CGT-GAG-AG**A**-**GA**3′	This study
F1731T	5′ACC-TGG-GTG-CGT-GAG-AG**G**-**AC**3′	This study
2365R	5′A-AGA-TGT-CGT-CGA-ACC-TCC-A3′	This study
F353T	5′CCT-GTT-CAT-TAC-GAC-GTG-GT**T**3′	This study
F353G	5′CCT-GTT-CAT-TAC-GAC-GTG-GT**G**3′	This study
R4	5′GGT-GCG-AGT-GAG-AAA-GCG-ACC3′	[30]

**Table 3 viruses-15-00959-t003:** RT-PCR profiles using F1731E/F1731T/2365R primers, deduced E vs. T/T’ pathotypes, and sequence validation of the molecular signature at position 1731–1733 for 49 RYMV isolates. ++, presence of a strong band, +, presence of a band, (+) presence of a feint band, − absence, ND, not determined.

F1731E/2365R Reactions	F1731T/2365R Reactions	Number of Isolates	Inferred Pathotypes	Sequence Validations
++	−	18	E	18/18
++	(+)	4		4/4
−	++	17	T/T’	17/17
(+)	++	9		9/9
+	+	4	ND	−

**Table 4 viruses-15-00959-t004:** RT-PCR profiles using F353T or F353G primers, deduced T/T’ pathotypes and sequence validation for 25 isolates. +, presence of a band, −, absence, ND, not determined.

F353G/R4 Reactions	F353T/R4 Reactions	Number of Isolates	Inferred Pathotypes	Phylogenetic Validations
+	−	18	T	18/18
−	+	7	T’	7/7
−	−	1	ND	−

## Data Availability

The sequence data used in this study are available on NCBI database (for details see Appendix A).

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
