# Peer review of "Molecular Tools to Infer Resistance-Breaking Abilities of Rice Yellow Mottle Virus Isolates"

_viruses, 2023, doi:10.3390/v15040959_

Round 1

Reviewer 1 Report

The manuscript “Molecular tools to infer resistance-breaking abilities of rice yellow mottle virus isolates” by Laurence and colleagues here describe a method based on the primer extension approach with detection of PCR product to provide a simple, fast and accessible tool for breeders and pathologists to infer the resistance-breaking abilities of RYMV isolates. The experiments were well designed and the results were sufficient illustrated and discussed. I think the manuscript is well written and I recommend this manuscript to be published in VIRUSES-Brief Report after revising some errors listed below:

1.     Please add more information about the RYMV genomic organization, taxonomic position in the introduction part.

2.     Line 141-142: please add space between number and unit. The same errors also present in the other text, please revise all of them.

3.     Figure 1 image resolution is two low

4.     There is only one polymorphism at position 350 was observed in 3 isolates (Cia, BF1, and CI63), I think only use G base to distinguish T/T’ pathotypes is not enough.

5.     Base on above point 4, forward primers F353T/F353G were designed with respectively T or G at their 3’extremity, this two forward primers have only one base different, I think it is hard to amplify their specific target in RT-PCR method, but Real time-quantitative RT PCR maybe more suitable. This two point can be added in discussion.

Author Response

Thank you for the evaluation of our manuscript. Please find below our responses.

1. Information about the RYMV genomic organization and its taxonomic position was added in the introduction part (lines 33-36 of the revised version) as suggested.

2. Space was added between number and unit (lines 149, 152, 153, 242, 254) as suggested.

3. Figure 1 resolution was increased as suggested.

4. As indicated in the text, in the figure 1 and the table 1, the polymorphism used to distinguish T/T' pathotypes is located at position 353 not 350.
The nucleotide T353 is strictly conserved in all the isolates that belong to the T' pathotype.

5. Despite the fact that the primers F353T/F353G have only one difference at their 3' extremity, we experimentally demonstrated that they amplify their specific target on 25/26 samples as indicated in the text.
We agree that Real time-quantitative RT-PCR may be suitable, however, this technology is not always accessible for breeders and pathologists in Africa.
These two points were added in the discussion section (lines 316-317, 339-340) as suggested. 

Reviewer 2 Report

The manuscript entitled “Molecular tools to infer resistance-breaking abilities of rice yellow mottle virus isolates" by Laurence Dossou et al designed specific RT-PCR primers to infer resistance-breaking ability of RYMV isolates based on viral protein genome-linked and polymorphism sites and was also tested and validated on 52 isolates representing the genetic diversity of RYMV.

1. The manuscript is not easy to follow. The article uses a large number of abbreviations, pathogenic isolate names, nucleotide numbers and sequence and primer names appear confusing, please improve if possible.

2. The correlation between RYMV lineage and RB abilities needs further elucidation.

3. Whether the RB abilities of the 52 isolates selected as representatives of genetic and pathogenic diversity of RYMV has been experimentally verified requires additional clarification.

4. Figure1 – The resolution of the image is unacceptable.

Author Response

Thank you for the evaluation of our manuscript. Please find below our responses.

1. Material and Methods '(lines 118-119 of the revised version) and Results (lines 182-184, 203, 224-225, 235-239 and 251-252, respectively) were modified to improve the clarity of the names of isolates, sequences and primers as suggested.

2. The introduction (lines 66-77, 92-96 and lines 106-108) was modified 
to better explain the correlation between RYMV lineage and RB abilities as suggested.

3. The text (lines 271, 293, 305-306, 312-313) was modified to be more explicit as suggested. The molecular tools designed in this study were experimentally validated on 52 isolates selected as representatives of the RYMV genetic and pathogenic diversity the RYMV isolates (pathotyped isolates are indicated by E, T, or T' in Figure 1). Using these tools, the polymorphisms at positions 353 and 1731-1733 (= VPg49) were determined without sequencing step and, based on these molecular signatures and published experimental data, pathotypes were inferred without greenhouse experiments.

4. Figure 1 resolution was increased as suggested.

Round 2
